# MicroRNA Expression Profile in Bone Marrow and Lymph Nodes in B-Cell Lymphomas

**DOI:** 10.3390/ijms242015082

**Published:** 2023-10-11

**Authors:** Yuliya A. Veryaskina, Sergei E. Titov, Igor B. Kovynev, Tatiana I. Pospelova, Sofya S. Fyodorova, Yana Yu. Shebunyaeva, Dina V. Sumenkova, Igor F. Zhimulev

**Affiliations:** 1Department of the Structure and Function of Chromosomes, Laboratory of Molecular Genetics, Institute of Molecular and Cellular Biology, SB RAS, 630090 Novosibirsk, Russia; titovse78@gmail.com (S.E.T.); zhimulev@mcb.nsc.ru (I.F.Z.); 2Laboratory of Gene Engineering, Institute of Cytology and Genetics, SB RAS, 630090 Novosibirsk, Russia; 3AO Vector-Best, 630117 Novosibirsk, Russia; 4Department of Therapy, Hematology and Transfusiology, Novosibirsk State Medical University, 630091 Novosibirsk, Russia; kovin_gem@mail.ru (I.B.K.); depart04@mail.ru (T.I.P.); soffka.tv@mail.ru (S.S.F.); jana.shebuniaeva@yandex.ru (Y.Y.S.); dinasumenkova@mail.ru (D.V.S.)

**Keywords:** microRNA, Hodgkin’s lymphomas, non-Hodgkin’s lymphomas, B-cell lymphoma, bone marrow, lymph nodes

## Abstract

Hodgkin’s lymphomas (HL) and the majority of non-Hodgkin’s lymphomas (NHL) derive from different stages of B-cell differentiation. MicroRNA (miRNA) expression profiles change during lymphopoiesis. Thus, miRNA expression analysis can be used as a reliable diagnostic tool to differentiate tumors. In addition, the identification of miRNA’s role in lymphopoiesis impairment is an important fundamental task. The aim of this study was to analyze unique miRNA expression profiles in different types of B-cell lymphomas. We analyzed the expression levels of miRNA-18a, -20a, -96, -182, -183, -26b, -34a, -148b, -9, -150, -451a, -23b, -141, and -128 in lymph nodes (LNs) in the following cancer samples: HL (n = 41), diffuse large B-cell lymphoma (DLBCL) (n = 51), mantle cell lymphoma (MCL) (n = 15), follicular lymphoma (FL) (n = 12), and lymphadenopathy (LA) (n = 37), as well as bone marrow (BM) samples: HL (n = 11), DLBCL (n = 42), MCL (n = 14), FL (n = 16), and non-cancerous blood diseases (NCBD) (n = 43). The real-time RT-PCR method was used for analysis. An increase in BM expression levels of miRNA-26b, -150, and -141 in MCL (*p* < 0.01) and a decrease in BM levels of the miR-183-96-182 cluster and miRNA-451a in DLBCL (*p* < 0.01) were observed in comparison to NCBD. We also obtained data on increased LN levels of the miR-183-96-182 cluster in MCL (*p* < 0.01) and miRNA-18a, miRNA-96, and miRNA-9 in FL (*p* < 0.01), as well as decreased LN expression of miRNA-150 in DLBCL (*p* < 0.01), and miRNA-182, miRNA-150, and miRNA-128 in HL (*p* < 0.01). We showed that miRNA expression profile differs between BM and LNs depending on the type of B-cell lymphoma. This can be due to the effect of the tumor microenvironment.

## 1. Introduction

According to the classification of hematolymphoid tumors, the term lymphoma covers a heterogeneous group of malignancies, divided into Hodgkin lymphomas (HL) and non-Hodgkin lymphomas (NHL), which differ in terms of clinical manifestations, morphological characteristics, and immunological and molecular phenotypes. HL and most NHL originate from different stages of B-cell differentiation [1]. The most common types of B-cell NHL are diffuse large B-cell lymphoma (DLBCL), which accounts for about 40% of adult cases, follicular lymphoma (FL; 15%), chronic lymphocytic leukemia/small lymphocytic lymphoma (CLL; 11%), marginal zone lymphoma (MZL; 7%), and mantle cell lymphoma (MCL; 6%) [2].

MicroRNA (miRNA) expression profiles change during lymphopoiesis [3]. Lai et al. showed that the miR-17-92 cluster participates in the regulation of early stages of lymphocyte development [4]. Other studies demonstrated that miRNA-181 and miRNA-34 are involved in the regulation of the pro-B to pre-B cell transition [5]. Souza et al. noted that miRNA-150, -146a, -155, -125b, -223, and -142, as well as the let-7 family of miRNAs, control B-cell differentiation [6]. In addition, miRNA-17, -24, -146, -155, -128, and -181 regulate early stages of hematopoiesis; miRNA-16, -103, and -107 can block the differentiation of late precursors; and miRNA-221, -222, and -223 control the terminal stages of hematopoiesis differentiation [7]. Apparently, miRNAs are regulators of normal lympho- and hematopoiesis, and their dysregulation can contribute to the development of B-cell lymphomas.

To date, a series of works aimed at studying the role of miRNAs in different subtypes of B-cell lymphomas has been published. Gao et al. showed that miRNA-19b, -193a, -370, -490, and -630 are differentially expressed in DLBCL and lymph node reactive hyperplasia [8]. Pan et al. noted the unique profiles of miRNA-9a and miRNA-17 expression in FL compared to reactive lymphatic nodes [9]. Furthermore, the analysis of miRNA-155, -21, and -26a expression levels allows for the differentiation between Burkitt lymphoma and primary DLBCL [10]. However, a number of studies report identical miRNA expression profiles in various B-cell lymphoma samples compared to normal lymph nodes, which indicates the presence of common molecular pathways in the development of these circulatory diseases. In particular, Kluiver et al. showed that miRNA-155 is overexpressed in HL and DLBCL [11]. Roehle et al. found that miRNA-210, -155, and -106a are upregulated, while miRNA-149 and miRNA-139 are downregulated in both DLBCL and FL [12]. Since lymphoid malignancies are classified based on the tumor cell origin and differentiation stage and characterized by a specific miRNA expression profile, the analysis of miRNA expression can be a reliable diagnostic tool, allowing one to differentiate tumors that cannot be classified using conventional diagnostic methods. Moreover, elucidation of the role of miRNAs in lymphopoietic disorders is an important fundamental task.

The aim of this study was to analyze unique miRNA expression profiles in different B-cell lymphomas.

## 2. Results

### 2.1. Comparative Analysis of miRNA Expression Levels between Tumor Samples and Non-Cancerous Blood Diseases

Using real-time RT-PCR, we analyzed the expression levels of miRNA-18a, -20a, -96, -182, -183, -26b, -34a, -148b, -9, -150, -451a, -23b, -141, and -128 in the LN and BM samples. A ˃3-fold difference in expression levels between the compared subgroups was considered significant. Threshold cycles Ct for miRNA-9 in BM exceeded cycle 35, and these data were excluded from the analysis.

#### 2.1.1. Diffuse Large B-Cell Lymphoma

We observed a statistically significant decrease in expression levels of miRNA-148b, -18a, -183, -20a, -26b, -451a, -96, and -182 in DLBCL BM samples compared to NCBD (*p* < 0.01) (Table 1). The most significant (>3-fold) decrease in expression in DLBCL BM was observed for the miRNA-183/96/182 cluster and miRNA-451a. A comparative analysis of miRNA expression in LN samples revealed a statistically significant downregulation of miRNA-26b, -148b, -150, -451a, and -128 in DLBCL compared to LA, while miRNA-150 expression was decreased >15-fold in DLBCL LNs (*p* < 0.01).

#### 2.1.2. Mantle Cell Lymphoma

We observed a statistically significant increase in BM levels of miRNA-148b, -18a, -150, -141, and -26b in MCL samples compared to NCBD (*p* < 0.01) (Table 2). The most significant (>3-fold) overexpression in MCL BM was found for miRNA-26b, -150, and -141. A comparative analysis of miRNA expression in LNs showed a statistically significant increase in the levels of miRNA-183, -20a, -26b, -96, and -182 in MCL compared to LA and a >8-fold increase in miRNA-183/96/182 cluster expression (*p* < 0.01).

#### 2.1.3. Follicular Lymphoma

We observed a statistically significant increase in expression levels of miRNA-26b, -451a, and miRNA-141 (*p* < 0.01) in BM samples in FL compared to NCBD (*p* < 0.01) (Table 3). A comparative analysis of miRNA expression in LN samples showed a statistically significant increase in the levels of miRNA-18a, -20a, -9, and -96, as well as a decrease in miRNA-26b in FL compared to LA, while the expression of miRNA-18a, -9, and -96 were found to be increased >3-fold in FL (*p* < 0.01).

#### 2.1.4. Hodgkin’s Lymphoma

A statistically significant decrease in miRNA-148b and miRNA-128 and a statistically significant increase in miRNA-18a and miRNA-26b levels was observed in BM samples in HL compared to NCBD (*p* < 0.01) (Table 4). A comparative analysis of miRNA expression in LN samples demonstrated a statistically significant decrease in miRNA-18a, -182, -150, -141, and -128 levels in HL compared to LA (*p* < 0.01). The most significant (>3-fold) decreases in expression levels in HL LNs were observed for miRNA-182, -150, and -128 (*p* < 0.01).

### 2.2. Bioinformatics Analysis of Pathways and Targets Involved in B-Cell Lymphomas

Using the miRPathDB2.0 resource, which is freely available at https://mpd.bioinf.uni-sb.de/ (accessed on 01.08.2023), for the analysis of target genes for miRNA-18a, -20a, -96, -182, -183, -26b, -34a, -148b, -9, -150, -451a, -23b, -141, and -128, we identified target genes participating in hematopoiesis and biological processes involving lymphocytes (Table 5).

Next, we conducted a bioinformatic analysis of cancer-associated pathways that are regulated by miRNA-18a, -20a, -96, -182, -183, -26b, -34a, -148b, -9, -150, -451a, -23b, -141, and -128. DIANA-mirPath v3.0 revealed 64 enriched pathways with an adjusted *p* < 0.05. We focused on 21 pathways directly associated with blood cancer. The results are summarized in Table 6.

Next, we conducted bioinformatic analysis of the miRNA target genes involved in the KEGG cancer pathways. The resulting data are reported using the miRNet 2.0 resource (Figure 1 and Figure 2). The miRNet is a miRNA network visual analytics platform (https://www.mirnet.ca/miRNet/home.xhtml). An inclusion criterion was the strongest evidence for miRNA–mRNA interaction according to the miRNaBase v. 9.0 based on analysis using three methods (qPCR, reporter assay, and Western blot). Furthermore, we decided to show DICER1, a key participant of miRNA processing, in figures, thus demonstrating that miRNAs regulating DICER1 expression significantly contributed to the expression profile of the entire set of miRNAs [13].

## 3. Discussion

The analysis of miRNA-18a, -20a, -96, -182, -183, -26b, -34a, -148b, -9, -150, -451a, -23b, -141, and -128 levels showed differential expression of the markers under study between different B-cell lymphoma subtypes and different expression profiles of these miRNAs between BM and LN samples for one B-cell lymphoma subtype.

We observed a statistically significant >3-fold increase in BM levels of miRNA-26b, -150, and -141 in MCL (*p* < 0.01), as well as a >3-fold decrease in BM expression of the miRNA-183/96/182 cluster and miRNA-451a in DLBCL (*p* < 0.01).

Hematopoiesis is a multistep process that involves the differentiation of hematopoietic stem cells (HSCs) in specific unipotent progenitor cells. Each differentiation stage is characterized by a specific set of genetic and epigenetic biomarkers [14,15]. MicroRNAs can contribute to the development of hematological tumors due to hematopoiesis impairment [16]. A series of ligand–receptor interactions that play a key role in HSC development have been identified to date: CXCL12 (SDF-1)/CXCR4, NOTCH1/JAG1 (Jagged-1), ANGPT1/ANGPT2/TEK (Tie-2), SPP1 (OPN)/CD44/integrin receptors, KITLG (SCF)/KIT (c-Kit), VEGFA (VEGF)/FLT1/KDR (VEGFR1/2), VCAM1/ITGA4 (VLA-4), and IGF1/IGF1R [17]. It has been shown that miR-34a regulates Notch1, Jagged1, VEGFA, and c-Kit expression [18,19,20]. He et al. note that miRNA-150 negatively regulates VEGF expression [21]. MiR-26b was reported to regulate Angpt1 expression in BM mesenchymal stromal cells [22]. The IGF1 signaling pathway regulates B cell lymphopoiesis, promoting differentiation of pro-B cells in pre-B cells [23]. Budzinska et al. showed that 3′UTR mRNA IGF-1R contains two putative miR-96-binding sites and two sites for miR-182 recruitment; they reported the miRNA-183/96/182 cluster as one of the regulators of hematopoiesis in normal conditions [24]. In addition, recent studies suggest an important role of the miRNA-183/96/182 cluster in oncogenesis, tumor progression, invasion, and metastasis of tumors of various origins [25,26,27].

A wide spectrum of blood cells, including B lymphocytes, is produced during hematopoiesis. The differentiation of B cells from lymphoid progenitors is regulated by a series of transcription factors, including PU.1, STAT5, E2A (E12 and E47), EBF, Pax-5, IKZFI, and FOXP1 [16]. There are data on miRNA-34a and miRNA-150 involvement in FOXP1 expression regulation and B-cell development [5,6].

We conducted bioinformatics analysis and a literature review, showing that miRNA-150, -18a, -20a, -23b, -34a, -451a, and -128 are involved in hematopoiesis regulation and, in particular, lymphocyte-related processes. Thus, it is apparent that expression of these miRNAs can contribute to development of hematological tumors.

Lymphomas are heterogeneous in clinical, morphological, and molecular aspects; at the same time, they have common traits. To date, a number of biomarkers with a diagnostic value for each B-cell lymphoma subtype have been determined: BCL2, BCL6, CD10, CD21, and STMN1 in FL; D1 and SOX11 in MCL; FOXP1 and MUM1 in ABC-DLBCL; CD10, BCL2, BCL6, LMO2, and GCET1 in GC-DLBCL; and CD30, CD15, EBV, BOB1, OCT2, and CD79a in HL [15]. An analysis of published data demonstrated that these biomarkers are targets for a number of miRNAs including those studied in this work. We showed that miRNAs are differentially expressed in different B-cell lymphoma subtypes.

We obtained data on the overexpression of miRNA-18a (*p* < 0.01), which is a miRNA-17-92 cluster member, and miRNA-9 (*p* < 0.01) in FL. Nowek et al. note that STMN1 regulates microtubule formation dynamics during cell cycle progression and serves as a target for miRNA-9 [28]. Arzuaga-Mendez et al. report an upregulation of miRNA-9 expression in the germinal center B lymphocyte [29]. This can explain the data we obtained, since FL belongs to GC lymphomas [30].

BCL6 and PRDM1 are considered the main regulators of germinal center formation and B-cell terminal differentiation, and changes in expression of these transcription factors are associated with lymphomagenesis [31]. Tsai et al. showed that BCL6 and BACH2 expression levels in lymphoblastoid cells are regulated by miRNA-34a, -148a, and -183 [32]. Lin et al. demonstrated that BCL-6 promotes miRNA-34a methylation, thus indicating the presence of more complicated regulatory pathways [33].

We showed that the miRNA-183/96/182 cluster is overexpressed in MCL and FL (*p* < 0.01). Experiments confirmed that SOX11 is a target for miRNA-182, and a decrease in miRNA-182 inhibits the growth of hepatocellular carcinoma cells and downregulates BCL2 expression [34]. Ye et al. report that BCL2 and HOXA9 are key miR-182 targets and that miR-182 is a tumor suppressor gene that inhibits self-renewal of leukemia stem cells [35]. Similar to our results, Di Lisio et al. noted the overexpression of miRNA-182 in MCL [36].

We demonstrated a >15-fold downregulation of miRNA-150 in LNs in DLBCL (*p* < 0.01). Musilova et al. note that a decrease in miRNA-150 level contributes to FL transformation to DLBCL due to an increase in FOXP1 level [37]. Another study reports that miRNA-17-92 cluster members and miRNA-150 are biomarkers that allow for the differentiating of FL from DLBCL [29]. Morris et al. conclude that LMO2 can inhibit miRNA-150 expression in B and T cells [38].

A study by Gibcus et al. shows that miRNA-150 expression is decreased in HL cells compared to NHL cells and normal GCB cells [39]. We also observed miRNA-150 downregulation in HL compared to LA (*p* < 0.01). However, unlike Gibcus et al., we found that miRNA-150 level was decreased >15-fold in LNs in DLBCL, which is one of the NHL subtypes (*p* < 0.01). Differences found between the results can be due to a different type of the sample used in the studies and also due to the fact that tumor microenvironment can affect the data obtained in the tumor sample compared to cancer cell lines.

We observed a decrease in the miRNA-183/96/182 cluster level in HL LNs. The latent membrane protein 1 (LMP1) is one of the main proteins synthesized by the Epstein–Barr virus (EBV) and expressed in Reed–Sternberg cells in EBV-associated HL [40]. Oussaief et al. demonstrated that LMP-1 downregulates the miR-183-96-182 cluster [41]. We did not perform EBV analysis in the HL LN sample in our work. We can only assume that part of the patients in the studied sample had EBV-associated HL; this fact can be a likely reason for decreased expression of the miRNA-183/96/182 cluster.

We showed that miRNA expression profile differs between BM and LNs depending on lymphoma subtype. This can be due to an effect of the tumor microenvironment. Recently, much attention has been paid to the study of the interaction between tumor cells and their microenvironment. This interaction is considered to play an important role in maintaining tumor cell proliferation, invasion, metastasis, and even resistance to therapy [42].

Despite a growing number of works aimed at the study of miRNA role as biomarkers of B-cell lymphoma, we are still far from introducing these markers into clinical practice. However, a number of miRNA-based diagnostic tests are available to date [43]. We demonstrated that different lymphoma subtypes have different miRNA expression profiles. Their ability to improve histological classification and stability in a biological sample make them ideal biomarkers. The results obtained in this study provide valuable data for understanding the pathogenesis of B-cell lymphoma. Further studies and clinical trials are required to confirm the prognostic and diagnostic potential of miRNAs.

## 4. Materials and Methods

### 4.1. Clinical Samples

A series of 176 formalin-fixed paraffin-embedded (FFPE) samples of lymph node (LN) HL (n = 41) and NHL, including DLBCL (n = 51), MCL (n = 15), and FL (n = 12), as well as non-tumoral samples (lymphadenopathy (LA) (n = 37)), were collected.

A total of 159 cytological samples were obtained by sternal puncture and aspiration biopsy of bone marrow (BM) from the posterior iliac spine. The study groups were HL (n = 11) and NHL including DLBCL (n = 42), MCL (n = 14), and FL (n = 16); the control group consisted of patients with non-cancerous blood diseases (NCBDs) (n = 43) (iron-deficiency anemia (n = 28), hemolytic anemia (n = 5), B12 deficiency anemia (n = 5), and immune thrombocytopenia (n = 5).

All the patients included in the study did not have signs of marrow damage, which was proven by sternal puncture and trephine biopsy followed by the morphological verification of bone marrow hematopoiesis and flow immunocytofluorimetry of cell aspirate (or sternal punctate) performed for each patient.

The characteristics of the groups are shown in Appendix A. Cytological and FFPE materials were obtained in compliance with Russian laws and regulations, written informed consent was obtained from each patient, and all the data were depersonalized. This study was approved by the ethics committee of the Novosibirsk State Medical University.

### 4.2. Isolation of Total RNA from Fine-Needle Aspiration Cytological Specimens

Each dried cytological specimen was washed in a microcentrifuge tube with three 200 µL portions of guanidine lysis buffer. Samples were vigorously mixed and incubated in a thermal shaker at 65 °C for 15 min. Next, an equal volume of isopropanol was added. The solution was thoroughly mixed and kept at room temperature for 5 min. After centrifugation at 14,000× *g* for 10 min, the supernatant was decanted, and the pellet was washed with 500 µL of 70% ethanol and 300 µL of acetone. The resulting RNA was dissolved in 290 µL of deionized water.

### 4.3. Isolation of Total RNA from FFPE

A total of 1 ml of mineral oil was added to a tube containing three 15 µm paraffin-embedded sections of lymph node tissue for deparaffinization. The tube was then vortexed for 10 s and incubated in a thermoshaker at 65 °C and 1300 rpm for 2 min. Next, samples were centrifuged at 13,000–15,000× *g* for 4 min. The supernatant was removed without disrupting the precipitate. A total of 1 ml of 96% ethanol was added to the precipitate, followed by vortexing for 10 s and centrifugation at 13,000–15,000× *g* for 4 min. The supernatant was removed without disrupting the precipitate, followed by the addition of 1 mL of 70% ethanol and centrifugation at 13,000–15,000× *g* for 2 min. The resulting precipitate was further used for nucleic acid isolation. In total, 600 µL of guanidine lysis buffer was added to each sample, and total RNA was isolated as described above.

### 4.4. Selection of miRNAs

MicroRNAs were selected based on literature data [44,45,46]. MicroRNAs with expression levels exceeding 100 units were included in our study based on these data. A total of 20 miRNAs were studied: miRNA-18a-5p, -20a-5p, -96-5p, -183-5p, -26b-5p, -34a-5p, -148b-3p, -9-5p, -150-5p, -451a, -23b-3p, -141-3p, and -128-3p. We added miRNA-182-5p to the analysis to assess the contribution of the miRNA-183/96/182 cluster to the pathogenesis of lymphomas. The geometric mean of Ct values of three miRNAs (-378-3p, -191-5p, and -103a-3p), which were selected based on our previous data [47], was used for normalization. The sequences of oligonucleotides for reverse transcription and PCR are presented in Appendix A. All oligonucleotides were synthesized by Vector-Best (Novosibirsk, Russia). Oligonucleotides were selected using the PrimerQuest tool (https://eu.idtdna.com/ (accessed on 1 June 2020)). The E value varied within the 92.5–99.7% range depending on the system used.

### 4.5. Reverse Transcription

For cDNA synthesis, reverse transcription was performed in a volume of 30 µL. The reaction mixture contained 3 µL of RNA sample and RT buffer solution with RT primer (Vector-Best, Novosibirsk, Russia). Total RNA concentration was within the range of 115–150 ng/µL; optical density 260/280 and 260/230 ratios were ≥1.9 and ≥1.5, respectively. The reaction mixture was incubated at 16 °C for 15 min and then at 42 °C for 15 min, followed by heat inactivation at 95 °C for 2 min.

### 4.6. Real-Time PCR

MicroRNA expression was assessed by real-time PCR using a CFX96 detection system (Bio-Rad Laboratories, Hercules, CA, USA). The total volume of each reaction mixture was 30 µL; the reaction mixture contained 3 µL of cDNA, 1× PCR buffer (Vector-Best, Russia), 0.5 µL of each primer, and 0.25 µL of a dual-labeled probe. The PCR protocol was as follows: incubation at 50 °C for 2 min and then pre-denaturation at 94 °C for 2 min, followed by 50 cycles of denaturation (94 °C for 10 s), annealing, and extension (60 °C for 20 s).

### 4.7. Statistical Analysis

The statistical analysis was performed using Statistica v13.1 software. The Mann–Whitney U test was used. *p* values < 0.01 were considered statistically significant.

## Figures and Tables

**Figure 1 ijms-24-15082-f001:**
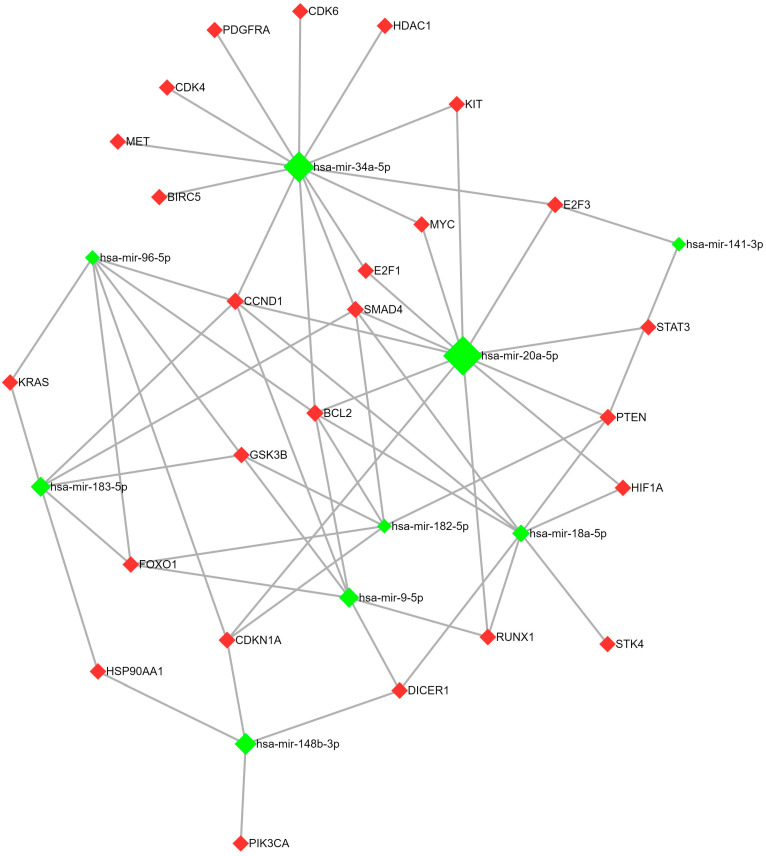
Target analysis of miRNA-18a, -20a, -96, -182, -183, -34a, -148b, -9, and -141 using miRnet 2.0. Green diamonds are microRNAs, red diamonds are their target genes.

**Figure 2 ijms-24-15082-f002:**
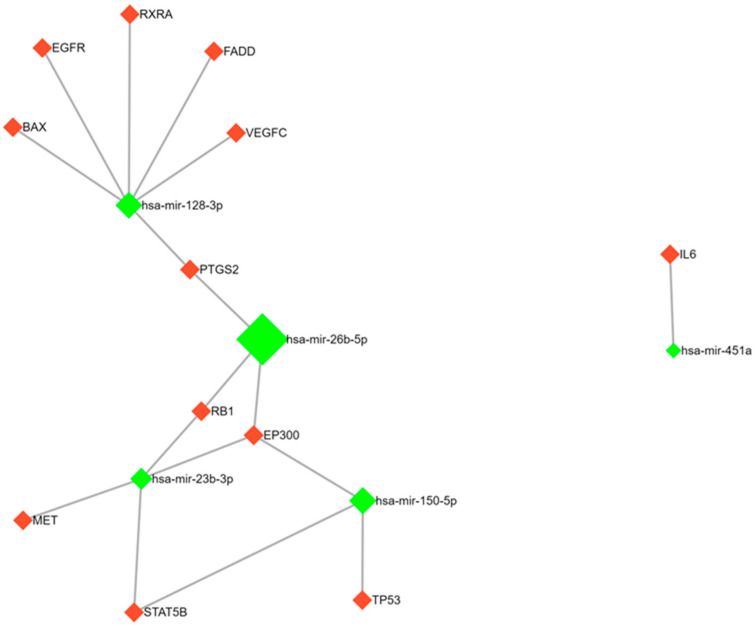
Target analysis of miRNA-26b, -150, -451a, -23b, and -128 using miRnet 2.0. Green diamonds are microRNAs, red diamonds are their target genes.

**Table 1 ijms-24-15082-t001:** Comparative analysis of miRNA expression levels between DLBCL and non-cancerous samples.

	Bone Marrow		Lymph Node	
	Fold Change	Adjusted *p*-Value	Fold Change	Adjusted *p*-Value
miR-18a	−1.46	2 × 10^−3^	1.48	NS
miR-20a	−2.08	2 × 10^−3^	−1.31	NS
miR-96	−3.10	2 × 10^−6^	1.48	NS
miR-182	−3.55	3 × 10^−6^	1.16	NS
miR-183	−3.41	1 × 10^−4^	1.36	NS
miR-26b	−1.93	1 × 10^−4^	−2.61	1 × 10^−5^
miR-34a	1.85	NS	−1.13	NS
miR-148b	−1.82	3 × 10^−8^	−1.87	1 × 10^−10^
miR-9	–	–	1.69	NS
miR-150	−1.77	NS	−15.66	1 × 10^−14^
miR-451a	−4.82	3 × 10^−5^	−2.79	1 × 10^−3^
miR-23b	−1.10	NS	−2.02	NS
miR-141	2.42	NS	−1.27	NS
miR-128	−1.01	NS	−1.72	5 × 10^−4^

NS—not significant.

**Table 2 ijms-24-15082-t002:** Comparative analysis of miRNA expression levels between MCL and non-cancerous samples.

	Bone Marrow		Lymph Node	
	Fold Change	Adjusted *p*-Value	Fold Change	Adjusted *p*-Value
miR-18a	1.44	3 × 10^−4^	−1.9	NS
miR-20a	1.49	NS	1.91	1 × 10^−2^
miR-96	−1.28	NS	10.2	1 × 10^−6^
miR-182	−1.47	NS	8.57	3 × 10^−8^
miR-183	−1.37	NS	8.68	2 × 10^−8^
miR-26b	4.24	3 × 10^−10^	2.52	4 × 10^−6^
miR-34a	3.5	NS	1	NS
miR-148b	1.34	9 × 10^−3^	1.13	NS
miR-9	–	–	1.72	NS
miR-150	3.44	1 × 10^−3^	−1.37	NS
miR-451a	1.55	NS	−1.52	NS
miR-23b	1.97	NS	−1.78	NS
miR-141	4.16	2 × 10^−4^	1.54	NS
miR-128	1.18	NS	−1.83	NS

NS—not significant.

**Table 3 ijms-24-15082-t003:** Comparative analysis of miRNA expression levels between FL and non-cancerous samples.

	Bone Marrow		Lymph Node	
	Fold Change	Adjusted *p*-Value	Fold Change	Adjusted *p*-Value
miR-18a	1.01	NS	8.77	6 × 10^−10^
miR-20a	1.16	NS	2.25	2 × 10^−3^
miR-96	−1.15	NS	3.69	1 × 10^−2^
miR-182	−1.16	NS	3.45	NS
miR-183	1.75	NS	2.77	NS
miR-26b	2.67	1 × 10^−9^	-2.26	7 × 10^−4^
miR-34a	−1.84	NS	2.23	NS
miR-148b	−1.24	NS	−1.53	NS
miR-9	–	–	4.12	5 × 10^−6^
miR-150	1.59	NS	−2.18	NS
miR-451a	2.21	1 × 10^−2^	−1.44	NS
miR-23b	−1.33	NS	1.11	NS
miR-141	1.72	2 × 10^−3^	1.05	NS
miR-128	−1.59	NS	−2.17	NS

NS—not significant.

**Table 4 ijms-24-15082-t004:** Comparative analysis of miRNA expression levels between HL and non-cancerous samples.

	Bone Marrow		Lymph Node	
	Fold Change	Adjusted *p*-Value	Fold Change	Adjusted *p*-Value
miR-18a	1.69	7 × 10^−4^	−1.72	1 × 10^−4^
miR-20a	−1.02	NS	−1.41	NS
miR-96	1.14	NS	−2.92	NS
miR-182	1.44	NS	−3.17	7 × 10^−3^
miR-183	−1.14	NS	−1.75	NS
miR-26b	2.21	5 × 10^−6^	−1.09	NS
miR-34a	−3.11	NS	1.51	NS
miR-148b	−2.24	1 × 10^−3^	−1.26	NS
miR-9	–	–	1.09	NS
miR-150	1.01	NS	−3.01	5 × 10^−5^
miR-451a	1.29	NS	−2.18	NS
miR-23b	1.04	NS	−1.44	NS
miR-141	−1.05	NS	−2.15	2 × 10^−4^
miR-128	−2.36	1 × 10^−6^	−3.22	1 × 10^−11^

NS—not significant.

**Table 5 ijms-24-15082-t005:** Biological pathways involved in hematopoiesis and lymphocyte differentiation with participation of the miRNAs under study. The list was generated using the miRPathDB 2.0 tool.

MiRNA	Pathway	*p*-Value	Targets
miR-34a	hematopoiesis	0.027	ATG5, AXL, BAX, BCL2, CDK6, CSF1R, DLL1, ERBB2, FOS, FOXP1, HDAC1, HMGB1, IFNB1, JAG1, KIT, KLF4, LEF1, MYB, MYC, NOTCH1, NOTCH2, SIRT1, TCF7, TP53, TREM2, WNT1, ZAP70
	microRNA pathway associated with chronic lymphocytic leukemia	0.002	BCL2, TP53, ZAP70
	lymphocyte activation	0.006	AKT1, ATG5, AXL, BAX, BCL2, CD24, CD44, CDK6, DLL1, ERBB2, FKBP1B, FLOT2, FOXP1, HMGB1, IFNB1, IMPDH2, KIT, LEF1, MYB, NOTCH2, PIK3CG, SRC, TCF7, TP53, ULBP2, WNT1, ZAP70
	lymphocyte differentiation	0.010	ATG5, AXL, BAX, BCL2, CDK6, DLL1, ERBB2, FOXP1, HMGB1, IFNB1, KIT, LEF1, MYB, NOTCH2, TCF7, TP53, WNT1, ZAP70
miR-18a	hematopoiesis	0.032	ATM, BCL2, FCGR2B, HIF1A, PIAS3, RUNX1, STK4, TGFBR2, TNFSF11
	lymphocyte activation	0.014	ATM, BCL2, FCGR2B, RUNX1, SDC4, SMAD3, TGFBR2, TNFAIP3, TNFSF11
	lymphocyte homeostasis	0.018	BCL2, HIF1A, TNFAIP3
	lymphocyte proliferation	0.030	ATM, BCL2, FCGR2B, SDC4, TGFBR2
miR-150	hematopoiesis	0.018	CCR6, CREB1, EP300, FLT3, MMP14, MYB, PRKCA, STAT1, STAT5B, TP53, VEGFA, ZEB1
	lymphocyte differentiation	0.018	CCR6, EP300, FLT3, MMP14, MYB, STAT5B, TP53, ZEB1
	lymphocyte activation	0.036	CCR6, EP300, FLT3, MMP14, MYB, P2RX7, STAT5B, TP53, ZEB1
	positive regulation of lymphocyte apoptotic process	0.047	P2RX7, TP53
miR-20a	hematopoiesis	0.039	APP, BCL2, EPAS1, HIF1A, KIT, MYC, PKNOX1, PPARG, RB1, REST, RUNX1, RUNX3, SMAD7, STAT3, TGFBR2, VEGFA
	microRNA pathway associated with chronic lymphocytic leukemia	0.016	BCL2, MCL1
miR-451a	lymphocyte homeostasis	0.016	AKT1, BCL2, MIF
	positive regulation of lymphocyte proliferation	0.040	BCL2, IL6, MIF
miR-23b	hematopoiesis	0.038	CA2, ETS1, HMGB2, NOTCH1, NOTCH2, PRDX3, PTK2B, RUNX2, SIRT1, TMEM64, ZEB1
miR-128	hematopoiesis	0.043	BAX, BMI1, CSF1, FADD, FBXW7, KLF4, LGALS3, MAPK14, MTOR, PIK3R1, SIRT1
	regulation of lymphocyte apoptotic process	0.008	BAX, FADD, LGALS3, PTEN

**Table 6 ijms-24-15082-t006:** Cancer-associated pathways in which the miRNAs in question are involved. The list was generated by DIANA-mirPath v3.0.

KEGG Pathway	Genes in the Pathway, Total	*p*-Value
**Cell biology**		
Cell cycle (hsa04110)	97	1 × 10^−9^
DNA replication (hsa03030)	26	9 × 10^−3^
RNA transport (hsa03013)	107	3 × 10^−2^
mRNA surveillance pathway (hsa03015)	61	3 × 10^−2^
Spliceosome (hsa03040)	84	4 × 10^−2^
**Cancer-associated pathways**		
Proteoglycans in cancer (hsa05205)	141	2 × 10^−10^
Chronic myeloid leukemia (hsa05220)	59	1 × 10^−6^
Transcriptional misregulation in cancer (hsa05202)	119	1 × 10^−5^
Pathways in cancer (hsa05200)	255	2 × 10^−5^
Central carbon metabolism in cancer (hsa05230)	49	4 × ^10−4^
Acute myeloid leukemia (hsa05221)	43	5 × 10^−4^
**Signaling pathway**		
p53 signaling pathway (hsa04115)	59	4 × 10^−7^
Hippo signaling pathway (hsa04390)	98	1 × 10^−5^
ErbB signaling pathway (hsa04012)	63	3 × 10^−5^
mTOR signaling pathway (hsa04150)	49	3 × 10^−5^
TGF-beta signaling pathway (hsa04350)	54	5 × 10^−4^
MAPK signaling pathway (hsa04010)	163	2 × 10^−3^
HIF-1 signaling pathway (hsa04066)	74	2 × 10^−3^
FoxO signaling pathway (hsa04068)	89	1 × 10^−2^
TNF signaling pathway (hsa04668)	73	2 × 10^−2^
AMPK signaling pathway (hsa04152)	84	4 × 10^−2^

## Data Availability

All data generated or analyzed during this study is available with the corresponding author (RSR) and will be provided on request.

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
