# Peer review of "MicroRNA Expression Profile in Bone Marrow and Lymph Nodes in B-Cell Lymphomas"

_ijms, 2023, doi:10.3390/ijms242015082_

Round 1
Reviewer 1 Report
The aim of the proposed study was to analyze unique microRNA expression profiles in different B-cell lymphomas. The authors used RT-PCR to evaluate the expression of several miRNAs in lymph nodes and bone marrow. Following this, the authors conducted bioinformatic analysis using an online available tool.
I have several comments and suggestions for the authors:
· The percentage of bone marrow involvement in a specific lymphoma can vary significantly. While reviewing the manuscript, there is no relevant discussion of this. Obviously, if you are extracting RNA from bone marrow with 5% lymphoma involvement versus 50% involvement, there will be significant changes in the results. In most cases of lymph node involvement, the typical appearance is total effacement of the node, meaning nearly 100% involvement. This can obviously be a major confounder. Could the authors explain how they analyzed the percentage of bone marrow involvement in each of their included lymphoma types?
· In Supplementary Figure 1, what did the author mean by the "bone marrow involvement" row under the bone marrow column for follicular lymphoma and mantle cell lymphoma? Aren't all the samples under the bone marrow column derived from patients with bone marrow involvement?
· In the Methods section, the authors wrote that "MicroRNAs were selected based on literature data. Sebestyén et al. quantified miRNAs in lymphoma samples using Nanostring technology”. However, this reference focused on CNS lymphomas, which are vastly different from systemic lymphomas in their clinical appearance. Can the authors explain their rationale for their miRNA selection process?"
· Why wasn't marginal cell lymphoma included in the study? Its prevalence is higher than mantle cell lymphoma.
· In the bioinformatic analysis, the authors used an online source to find gene targets for their miRNAs of interest. However, since these miRNAs were chosen from another paper that focused on CNS lymphoma, it is not surprising that the gene targets from that list will identify target genes participating in hematopoiesis and biological processes involving lymphocytes. Could the authors refine their bioinformatic analysis and provide predictions for pathways/gene targets related to their study (involvement of lymph nodes/bone marrow)? This should involve more than using a single open source online tool.
· Figure 1 resolution is too low- I can’t evaluate it
· Please change “lymphatic” to “lymph” in the title
· Corrected p-value should be Adjusted p-value
Reviewer 2 Report
This research analyzed a large series of lymphoma cases for the search of several miRNAs. The analysis was done by RT-PCR. It is a complex and difficult analysis regarding the interpretation of the data as the function of these miRNAs is not clear or have multiple effects. I am not sure if the authors should apply multiple comparison analysis or any type of multivariate analysis; the significance level could be reduce to 0.01 to highlight the most relevant markers, or try principal component analysis. The authors may consult with bioinformatician.
The work is worth publishing, the manuscript is easy to read, and to understand.
Additional comments:
(1) In the abstract. Line 23. I think it is just fine the call it "follicular lymphoma".
(2) Abstract, line 24. Regarding "lymphadenopathy". As I understand, are these samples reactive lymphoid tissue?
(3) Also in line 24. Regarding the bone marrow samples. I understand that there is a mixture of neoplastic and benign conditions?
(4) Regarding the abstract. Are the lymph node and the bone marrow samples from different patients? Are any paired sample?
(5) Line 39. The WHO-HAEM5 classification is still in beta/proposal version. Please also cite the WHO revised 4th edition, and the The International Consensus Classification of Mature Lymphoid Neoplasms: a report from the Clinical Advisory Committee https://doi.org/10.1182/blood.2022015851
(6) Line 46. Could you please confirm that CLL/SLL is not more frequent than the other mature B cell neoplasms?
(7) In lines 48 to 57 the function of several miRNAs is shown, but, are these the ones that are being analyzed?
(8) Line 76. What is the role of the miRNAs that are being analyzed in this project, and why were selected? Are they expressed by the neoplastic b-lymphocytes of by the cells of the microenvironment?
(9) Line 83. A >3 fold difference in expression level was used as cutoff. But what was the reference group? The reactive lymphoid tissue or any lymphoma subtypes with the lowest expression?
(10) In the different tables, is the comparison always lymph node hematological neoplasia vs reactive lymph node, and neoplasia bone marrow vs. reactive bone marrow?
(11) In Hodgkin lymphoma, was the percentage of HRS cells quantified? Did all samples have the same numbers of HRC cells? Could the miRNAs come from the inflammatory microenvironment (for example, macrophages, T cells, eosinophils, etc.?)
(12) In Figure 1. The name of the different pathways is too small and cannot be read.
Round 2
Reviewer 1 Report
“However, our study includes cases of malignant lymphomas”-The authors responded that their study includes cases of malignant lymphomas with proven absence of leukemia. There is no connection between leukemia (presence of blasts in peripheral blood and bone marrow) and the involvement of BM with NHL
“Sebestyén et al. quantified miRNAs in lymphoma samples using Nanostring technology”- this sentence is still present in the manuscript, albeit the authors changed the adjacent reference and now it is addressing 3 papers. The authors should revise the text, let the reader understand exactly how their miRNA selection process was conducted. Were miRNAs appearing in just 1 of the 3 references were included? a combination of all 3?
The manuscript currently has 6 tables and 0 figures. The authors should include at least one figure for better visualization of their results. One possible option is to have a gene target-miRNA map via Cytoscape
Round 3
Reviewer 1 Report
The authors addressed all my concerns.